# Prevalence of Blood Types and Alloantibodies of the AB Blood Group System in Non-Pedigree Cats from Northern (Lombardy) and Southern (Sicily) Italy

**DOI:** 10.3390/ani10071129

**Published:** 2020-07-03

**Authors:** Eva Spada, Roberta Perego, Luciana Baggiani, Elisabetta Salatino, Vito Priolo, Cyndi Mangano, Maria Grazia Pennisi, Daniela Proverbio

**Affiliations:** 1Veterinary Transfusion Research Laboratory (REVLab), Department of Veterinary Medicine (DIMEVET), University of Milan, 20133 Milan, Italy; luciana.baggiani@unimi.it (L.B.); elisabetta.salatino@hotmail.com (E.S.); daniela.proverbio@unimi.it (D.P.); 2Department of Veterinary Sciences, University of Messina, 98168 Messina, Italy; vitopriolo@live.it (V.P.); cyndi_m@hotmail.it (C.M.); mariagrazia.pennisi@unime.it (M.G.P.)

**Keywords:** alloantibodies, blood type, feline, Italy, neonatal isoerythrolysis, transfusion reaction

## Abstract

**Simple Summary:**

The most important blood group system in cats is the AB, in which cats are classified as type-A, B or AB. Cats have antibodies against the blood type they do not possess, called alloantibodies. The aims of this study were to update blood type prevalence in cats from Northern Italy and study for the first time the blood type in cats from an insular region of Southern Italy, Sicily; to detect alloantibodies in these feline populations; to compare results with previous studies performed in Italy and between regions in Northern and Southern Italy. Cats from Southern Italy had the highest prevalence of type-B and type-AB, and the lowest prevalence of type-A blood in Italy. In particular, type-AB prevalence was higher than all previous reports in non-pedigree cats in Europe and the Italian prevalence of anti-type-B alloantibodies in type-A cats was the lowest reported worldwide. These results highlight the usefulness of regional studies to report different prevalences in feline blood types. Compatibility tests such as blood typing and cross matching must be considered fundamental in cats of any origin to ensure safe and efficient blood transfusion and to prevent neonatal isoerythrolysis.

**Abstract:**

The aims of this study were to determine the prevalence of A, B and AB blood types and alloantibodies in non-pedigree cats from two regions, one in Northern and one in Southern Italy (Lombardy and Sicily, respectively). A total of 448 samples (52.0% from Northern and 48.0% from Southern Italy) were blood typed. The prevalence of A, B and AB blood types in northern and southern cats were 91.0%, 5.2%, 3.8%, and 77.2%, 12.1% and 10.7%, respectively. The prevalence of type-A blood in southern cats was significantly lower (*p* = 0.0001) than in northern cats, while type-B and AB blood were significantly higher (*p* = 0.0085 and *p* = 0.0051, respectively) in Southern compared to Northern Italian cats. Alloantibodies against type-A blood were found in 94.1% of type-B cats, 11.2% of type-A cats had alloantibodies against type-B blood, while no type-AB cats had alloantibodies with no significant difference between the two Italian populations. Type-AB prevalence in non-pedigree cats in Southern Italy was the highest reported in Europe. Italian type-A cats had the lowest worldwide prevalence of alloantibodies against type-B blood. These results highlight the usefulness of regional studies to report different prevalences in feline blood types and reinforce the importance of blood typing cats before transfusions and mating.

## 1. Introduction

The most clinically important and well-studied blood group system in cats is the AB system in which cats are classified as types A, B or AB based on the molecular nature of the blood group antigen gangliosides [1,2]. Cats have antibodies against red blood cells (RBCs) of different blood types, called alloantibodies. These are naturally-occurring—requiring no prior sensitization by transfusion or pregnancy [3]. All type-B cats have been reported to have alloantibodies against type-A erythrocytes, approximately one third of type-A cats have low-titer macroscopic anti-B agglutinating alloantibodies, but no alloantibodies have been found in type-AB cats [3]. Alloantibodies against different blood types in the cat are responsible for transfusion reactions, even when a blood recipient receives a first mismatched blood transfusion [4]. A marked acute haemolytic transfusion reaction generally occurs [5] (which may be fatal) if a type-B recipient receives type-A or AB blood. Less severe haemolytic transfusion reactions lead to the premature destruction of transfused red cells in type-A cats receiving type-B or AB blood. Type-AB cats may have transfusion reactions when receiving type-B and A blood, due to anti-A or anti-B alloantibodies in the donor plasma [6]. The anti-A alloantibodies of type-B queens are also responsible for the development of neonatal isoerythrolysis in type-A and AB kittens following the absorption of colostral anti-A alloantibodies within 24 h of birth [7,8].

Type-A blood is the most prevalent blood type worldwide. Type-B and type-AB blood types were reported to be rare in initial studies of non-pedigree cats [9,10,11,12,13,14]. However, breed and geographical variations of feline blood type prevalence have been shown. In European countries, the prevalence of type-B cats in non-pedigree cat populations can vary from 0 to 33% [13,15,16,17,18]. A large study highlighted a frequency of approximately 1.5% of type-AB in the European cat population, with the majority of type-AB cats being Ragdoll and European domestic shorthair (DSH) breeds [19]. Studies have demonstrated variation in the distribution of blood types in non-pedigree cats, not only among countries, but also within countries [9,14,15,17,20,21,22,23,24,25,26,27,28,29]. Similarly, alloantibody prevalence may show geographical variation for both blood type-A and B cats [17,30,31].

Previous studies on blood type prevalence in cats from Italy, have investigated different populations of cats from Northern [12,32] and Central Italy [33,34,35]. However, to the author’s knowledge, the prevalence of alloantibodies against foreign AB blood group antigens system has not been investigated in cats in Italy and there are no data about the prevalence of feline blood types in the South of the country. The aims of this study were therefore to: (I) update blood type prevalence in cats from Northern Italy (Lombardy region) and study for the first time the blood type in a population of cats from a region in Southern Italy, Sicily, the largest Italian island; (II) detect alloantibodies in these feline populations; III) compare results with previous studies performed in Italy and between cats from these Northern and Southern Italian regions. As type-A blood is the most prevalent in all non-pedigree cats worldwide, the hypothesis of the study was that cats from Sicily may have a different prevalence of the most rare blood types-B and type-AB and a different prevalence of alloantibodies to northern cats. The rationale being that these cats were from an island where a small genetic pool combined with inbreeding could have increased the number of cats with rare blood types over time.

## 2. Materials and Methods

### 2.1. Samples and Population

Ethylenediamine tetra-acetic acid (EDTA)-anticoagulated surplus feline blood samples collected for various clinical reasons at the University of Milan (Northern Italy) and at the University of Messina (Southern Italy) were randomly included in the study between November 2014 and December 2015. Samples from all types of Italian cats were included in this study, i.e., samples collected from: (I) owned cats presented for routine check-ups before neutering, or for diagnostic purposes in the case of unhealthy cats, (II) shelter cats before surgery for neutering, (III) stray colony cats. This last group comprised stray cats that live in colonies in urban and rural areas of Italian cities and are captured for a trap, neuter, and release (TNR) sterilization program as part of a national program to control stray pet populations under Italian National Law (law no. 281/1991). Blood sampling in all cats was originally performed by veterinarians for the patient’s benefit and diagnostic purposes only and only surplus blood was used for this study. Based on both the University of Milan’s and the University of Messina’s animal use regulations, formal ethical approval was therefore not needed as cats were sampled for diagnostic purpose and informed consent was given by the owners, the director of the shelters and the legal representative of the feline colonies for use of surplus blood samples and data for scientific purposes. 

All analyses were performed at the Veterinary Transfusion Research Laboratory (REVLab) of the University of Milan where the Northern Italian cats were sampled. Blood samples from cats in Southern Italy were stored at 4–6 °C after collection and were shipped refrigerated to the REVLab where they were analyzed within 48 h of arrival. 

The sample size was estimated assuming a prevalence of 2.1% of type-AB cats based on results obtained in previous prevalence studies in the Northern Italian feline population [12]. Therefore, we aimed to enroll at least 210 cats (95% confidence interval, 1% error) at each site (Northern and Southern Italy). For each blood sample, the cat’s age, sex, and origin (owned, shelter, and stray colony) were recorded. Postcode was used to determine the province of origin for cats.

### 2.2. Blood Typing

Blood typing was performed using the tube agglutination technique (TUBE) as previously described [27,36,37]. Briefly, to prepare a RBC suspension, 0.5 mL of EDTA-blood was centrifuged at 1600× *g* for 10 min at room temperature. Plasma was collected into a separate tube and was stored at −20 °C until tested for alloantibodies screening. The RBC pellet was washed by adding 2.5 mL of isotonic 0.9% NaCl solution. Following centrifugation at 1000× *g* for 1 min, the supernatant was removed, and the pellet was resuspended and then recentrifuged twice and finally reconstituted to a 5% RBC suspension. Polyclonal antibodies contained in type-B cat plasma (obtained from a type-B blood donor, collected with CPD anticoagulant at 1:7 ratio, stored at −20 °C) were used as primary reagents for the detection of type A red cell antigens. *Triticum vulgaris* lectin (Sigma-Aldrich, 8 μg/mL) was used for the detection of type-B RBCs antigens as this lectin binds to the NeuAc terminal of the type-B ganglioside and therefore strongly agglutinates feline type-B RBCs, but does not agglutinate type-A RBCs [36]. A 0.9% NaCl solution (saline solution) was used as a negative control. In 3 glass tubes (PYREX^®^ Tube Borosilicate glass 12 × 75 mm, Coming Inc., New York, NY, USA), 50 μL of 5% RBC suspension was mixed with 100 μL of type-B plasma (anti-A reagent, tube A), 100 μL of *Triticum vulgaris* lectin solution (anti-B reagent, tube B), or 100 μL of saline (control reagent, tube C), respectively. These mixtures were incubated at room temperature for 15 min before centrifugation for 15 s at 1000× *g*. Tubes were then gently agitated, and blood type was recorded for the tube where macroscopic agglutination was present (Figure 1). Type-B and AB samples were confirmed by back typing technique as previously described [12,38].

As samples collected in Sicily were shipped refrigerated to northern laboratory to be tested, we preliminarily assessed the stability of samples by typing one sample of each blood type, stored at 4 ± 2 °C, using TUBE after 24, 48, 72, and 96 h, and after 1, 2, 3, and 4 weeks of storage. Each sample stored at 4 ± 2 °C for up 1 month was correctly blood typed by TUBE.

### 2.3. Alloantibody Screening

Alloantibody screening is a modified major crossmatch, using RBCs of a known blood group, thus screening for naturally alloantibodies against that group in the recipient. Alloantibody screening was performed as previously described on frozen plasma samples [38,39,40]. Briefly, in a glass tube 25 μL of the type-A RBC suspension (5% in 0.9% NaCl) was added to 50 μL of type-B plasma. The same was done with type-B RBCs and type-A plasma and with type-A and type-B RBCs with type-AB plasma. The solution was gently mixed, incubated for 15 min at room temperature and then centrifuged for 15 s at 1000× *g*. Agglutination was evaluated by gently agitating the tube to resuspend the non-agglutinating RBCs in the cell button. The presence of alloantibodies against RBCs of a different blood type was identified if macroscopic agglutination was present, as in the previous description of the tube technique for blood typing. In addition, where agglutination was not macroscopically visible, microscopic agglutination was evaluated at ×40x. Any RBC agglutination, macroscopic and/or microscopic, but not rouleaux presence alone, was considered a positive result for presence of alloantibodies in the tested plasma sample. 

### 2.4. Statistical Analysis 

Prevalence of blood types and alloantibodies were calculated as the proportion of samples testing positive divided by the total number of tested samples and was presented with percentage and 95% confidence interval (95% CI). Data are shown as mean ± standard deviation (SD) or median and range (min-max) based on data distribution. To test the significance between age in Northern and Southern cats Mann–Whitney *U* test or Student’s *t*-test for independent samples were used. Comparisons between categorical data were made using Fisher’s exact test or χ^2^ test. Alloantibody prevalences between northern and southern cats were compared using comparison of proportions. A *p* value < 0.05 was considered significant. All statistical analyses were performed using a statistical software package (MedCalc software, version 19.1.3).

## 3. Results

### 3.1. Demographic Data and Blood Type 

A total of 448 blood samples were blood typed, 233/448 (52.0%) and 215/448 (48.0%) from the Lombardy region (Northern Italy) and from Sicily (Southern Italy), respectively. Samples were blood typed after a mean time of 12 days (±SD 6 days, range 2–28 days). Age was not recorded in 42 cats (36 from Northern Italy, 6 from Southern Italy). The median age in Northern Italian cats (median 3 yrs, range 0.3–18 yrs) was significantly higher than in Southern ones (median 2 yrs, range 0.3–19 yrs) (*p* < 0.0001). Sex was not reported in 20 cats (all from Northern Italy) and no significant difference in male and female prevalence was found between northern (male *n* = 102, female *n* = 111) and southern (male *n* = 96, female *n* = 119) cats (*p* = 0.5025). A significantly higher number of owned cats were included in the northern (*n* = 120) population than from Southern Italy (*n* = 64) (*p* < 0.0001) and a higher number of colony stray cats were from Southern (*n* = 115) rather than from Northern Italy (*n* = 79) (*p* < 0.0001).

Prevalence of blood types is reported in Table 1. The B and AB blood type were significantly more common (*p* = 0.0085 and *p* = 0.0051, respectively) in non-pedigree DSH cats from Southern Italy and type-A less common (*p* = 0.0001) compared to cats from Northern Italy.

No statistical significant association was found between blood types and sex and origin (owned, shelter or colony cats) variables, either in the total feline population nor Northern and Southern Italian feline populations. Based on owner, shelter or colony postcode, cats with blood type-A, B or AB were not geographically clustered among the provinces from which cats were recruited both in Southern and in Northern Italy. 

The blood type prevalence in non-pedigree DSH cats previously reported in Italy and the results of the current study are reported in Table 2.

### 3.2. Alloantibodies 

Alloantibodies screening were performed on frozen plasma samples within six months from RBCs separation. Due to insufficient residual volumes of plasma samples, alloantibodies were tested in only 327/448 (72.9%) samples: 196/233 (84.1%) from Northern Italy and 131/215 (60.9%) from Southern Italy (Table 3). In particular alloantibodies were tested in 268/378 (70.9%) type-A samples, 34/38 (89.5%) type-B samples and 25/32 (78.1%) type-AB samples. 

Almost all (32/34, 94.1%) blood type-B samples had alloantibodies against type-A RBCs and all 32/32 (100.0%) showing a macroscopic agglutination reaction. Two type-B plasma samples did not agglutinate with type-A RBCs: a two month old owned DSH female kitten with panleukopenia from Northern Italy, and a 6 year old owned DSH male cat from Southern Italy with a wound on the chin. 

Only 30/268 (11.2%) type-A cats had alloantibodies against type-B RBCs, 13.4% (4/30) of these showed a macroscopic agglutination reaction while the majority (26/30, 86.7%) had only a microscopic agglutination. 

No type-AB tested samples (0/25) had alloantibodies against type-A or type-B RBCs. No significant differences in alloantibody presence were detected between southern and northern cats. 

## 4. Discussion

This epidemiological study confirmed the prevalence of type-A blood at a worldwide high level in both investigated areas of Northern (Lombardy) and Southern (Sicily) Italy compared to studies in non-pedigree cats in Europe [9,14,25,41,42,43,44], America [8,10,23,29,37,45] Australia [27,28] and Asia [13,22,46,47,48]. Although the prevalence of type-A cats in Southern Italy was significantly lower than in Northern Italy, blood type-A is more prevalent in southern cats as has previously been reported in non-pedigree cats from Northern and Central Italy [12,32,33,34], in Ragdolls in Northern Italy [38] and in Maine Coons in Central Italy [35]. 

The prevalence of blood type-B in non-pedigree cat populations in Europe varies from 0 to 30% [9,17,24,26,41,42,49] with some countries, such as Greece and England, showing the highest prevalences of 20% and 30.5%, respectively [15,17,18]. Significant difference in prevalence of blood type-B was also found between regions within countries, such as in non-pedigree cats in the UK, where prevalence ranges from 7.9% in Northern to 30.5% in Southern UK [15,17]. In the current study the type-B prevalence was significantly different between the two investigated areas, with a higher value (12.1%) in southern cats. Type-B prevalence of southern cats was also higher compared to previous studies of non-pedigree DSH cats in Northern Italy, where it ranged from 2.1 to 5.7% [12,32], but it was similar to type-B prevalence previously reported (11.2%) in 401 non-pedigree cats from Central Italy [26].

We highlight a high prevalence (10.7%) of type-AB cats in the studied region of Southern Italy, which is significantly higher than the 3.8% in cats from the region from Northern Italy. Prevalence of type-AB blood in previous Italian studies varied in non-pedigree cats from 2.1% [12] to 5.7% [32] in Northern Italy, with the highest prevalence (18%) found only in pedigree Italian Ragdoll cats in catteries located in Northern Italy [38]. Thus, the prevalence of type-AB in non-pedigree cats in Northern Italy was between that found in two previous studies. However, no type-AB cats were found in early studies in Central Italy [26,33]. Type-AB blood is rare (≤1%) in non-pedigree cats in most studies [8,10,11,14,16,27,29,37,38,41,42,43,47,48,50,51]. The prevalence of type-AB blood in non-pedigree cats in Southern Italy was therefore very high and comparable only with that reported in non-pedigree cat populations from Japan (9.7%) and Israel (14.5%) [21,22]. Similarly to findings in our study, different prevalence of type-AB cats within countries have previously been reported, as seen between Northern (6.3%) and Central (0.4%) Portugal [14,25]. The high prevalence of type-AB cats in the Southern Italy cat population may reflect the relatively small gene pool from which this population has developed. Sicily is separated from the mainland by a marine natural barrier that prevents free movement of terrestrial fauna and the non-pedigree cat population is isolated from neighboring areas. These circumstances may have enhanced inbreeding in Sicilian domestic cats as is recognized in wild cats [52]. This might be particularly important in colony stray cats which are free roaming and in which the level of inbreeding may be more pronounced than in owned cats. Therefore, differences in sampling feline populations between Northern and Southern Italy could be another possible explanations for the high prevalence of the rare blood types B and AB in the southern population. However, the statistical analysis found no significant association between blood type and the different origins of the tested cats. 

When considering the prevalence of blood type in cats, the blood typing method used must be considered. While TUBE is considered the gold standard technique in feline blood typing [39,40], questions have been raised about the accuracy of the card-typing technique, with type-AB cats consistently producing weak agglutination [12,40]. With agglutination on card-typing tests, type-AB cats might be misinterpreted as type-B cats because of possible lower grade agglutination in the anti-A reagent well and a stronger agglutination on the anti-B reagent well [12,40,53]. This can result in misclassification, in particular if type-B plasma was not back typed to screen for the presence of anti-*A* alloantibodies or type-B samples were not tested with another method. This may explain variations in the prevalence of type-B and type-AB cats in the older epidemiological studies. There is less misclassification in more recent studies using the immunochromatographic method which has a higher sensitivity and specificity than the card-typing test for typing both type-B and AB samples [53]. 

To the best of our knowledge, this study is the first time sample stability has been tested for blood typing by TUBE. Based on our results, an additional advantage of TUBE is that the samples can be typed up to one month after collection (following storage at 4 ± 2 °C), and this could be useful in epidemiological studies and when working with samples collected from distant geographical regions, as in our study. 

Geographical variation has also been reported for alloantibody prevalence, and prevalence of anti-B alloantibodies in blood type-A non-pedigree cats ranged from 12.5% in Portugal [25] to 87.6% in Turkey [30]. Interestingly, a lower proportion of type-A blood (11.2%) in Italian cats showed anti-B alloantibodies compared with other studies. Most type-A cats positive for anti-B alloantibodies (86.7%) have a weak reaction due to low antibody titers as previously reported [3,21,27,37,54]. Other than geographical variations, testing methods and differences in individuals to accurately recognize microscopic agglutination may be influence the results of alloantibodies screening in previous studies. In addition, it is possible that the delay in separating plasma from RBCs in some samples, and the long storage time before analysis, could have influenced the results of alloantibody screening in our study, reducing, and thus limiting, recognition of the weak RBC agglutination reactions characteristic of type-A alloantibodies.

Two type-B cats had no alloantibodies. One was a 2-month-old kitten with panleukopenia. Usually type-B kittens develop their own anti-A alloantibodies at 6–8 weeks of age regardless of exposure to type-A erythrocytes. At 3 months of age, their plasma anti-A titers reach levels comparable with that found in adult type-B cats [3]. Young age could therefore explain the absence of alloantibodies against type-A. In addition, panleukopenia might have impaired the kitten’s immune response impacting alloantibody production and level. For the other type-B cat we have no additional information to explain the absence of alloantibodies against type-A RBCs; however, previous studies have shown that very few type-B cats lack alloantibodies [21,27,55]. 

As expected, our study confirmed that type-AB cats do not have type-A or type-B alloantibodies as previously shown [3,27,30].

This study had a number of limitations. The most important limitation is the difference in populations between the samples from Northern and Southern Italy. The two feline populations were randomly sampled, but the cats from Northern Italy were older and more likely to be owned than those in Southern Italy, while more colony stray cats were blood typed from Southern Italy. However, these variables were not significantly associated with blood type or alloantibody prevalence [12,16,18,21,22,24,27,32,37,38,47,49,54]. The fact that more colony stray cats were blood typed in Southern Italy than in Northern Italy could be a bias that might alter the prevalence of blood types—in these cats, the level of inbreeding was not known and some of these cats might have been related. A high level of inbreeding in these cats would explain the high prevalence of the rare blood type-B and AB found in Southern Italy cats in this study, even no statistical association was found between blood type and origin of the feline population tested. Due to the limited quantity of plasma samples, alloantibody screening was not always possible. Finally, another clinically relevant red cell antigen called Mik has been identified in cats [56]. This blood type was not investigated in this study since blood typing reagents for this blood type are not readily available.

## 5. Conclusions

We provide for the first time information on the prevalence of blood type in cats from a region of Southern Italy, Sicily and of prevalence of alloantibodies in Italian cats. Cats from Southern Italy had the highest prevalence of type-B and type-AB and lowest prevalence of type-A blood in Italy. In particular, type-AB prevalence was higher than in all previous reports in non-pedigree cats in Europe and the prevalence of anti-type-B alloantibodies in type-A Italian cats was the lowest reported worldwide. These results highlight that regional studies are useful to report different prevalences in feline blood types and alloantibodies. Compatibility tests such as blood typing and cross matching must be considered fundamental in cats of any origin to ensure safe and efficient blood transfusion and to prevent neonatal isoerythrolysis.

## Figures and Tables

**Figure 1 animals-10-01129-f001:**
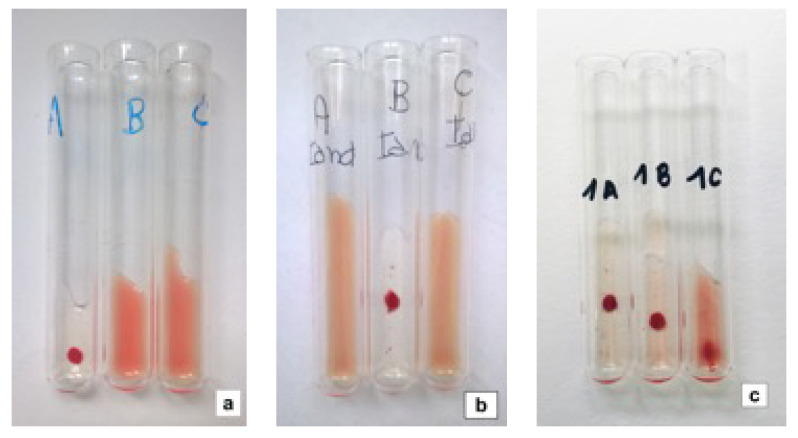
Type-A (**a**), B (**b**) and AB (**c**) feline blood samples analyzed with the tube agglutination technique. Blood type was determined by the presence of agglutination in the corresponding tube assigned as A or/and B and absence of agglutination in C (control) tube.

**Table 1 animals-10-01129-t001:** Blood type prevalence of AB blood group system in cats from regions of Northern (Lombardy) and Southern (Sicily) Italy.

Variables		*n* Positive/*n* Tested (Percentage: 95% Confidence Interval)			*p* Value
		Whole Population	Northern Italy	Southern Italy	
**Blood type**	**A**	378/448 (84.4%: 76.0–93.3)	212/233 (91.0%: 79.1–104.9)	166/215 (77.2%: 65.9–89.8)	**0.0001**
	**B**	38/448 (8.5%: 6.0–11.6)	12/233 (5.2%: 2.6–8.9)	26/215 (12.1%: 7.9–17.7)	**0.0085**
	**AB**	32/448 (7.1%: 4.8–10.0)	9/233 (3.8%: 1.7–7.3)	23/215 (10.7%: 6.7–16.0)	**0.0051**

Bold *p* values are statistically significant.

**Table 2 animals-10-01129-t002:** Prevalence of blood types of AB blood group system in non-pedigree DSH cats in epidemiological studies previously performed in Italy and compared with the results of the current study.

Country	Region	Blood Type			Blood Typing Method
		A	B	AB	
**Northern Italy**	**Lombardy** (*n* = 140) **[12]**	90.7%	7.1%	2.1%	Gel column agglutination
	**Lombardy** (*n* = 233) (current study)	91.0%	5.2%	3.8%	Tube agglutination
	**Piedmont** (*n* = 122) **[32]**	86.9%	7.4%	5.7%	Card agglutination
**Central Italy**	**Tuscany** (*n* = 401) **[26]**	88.8%	11.2%	0.0%	Microplate agglutination
**Southern Italy**	**Sicily** (*n* = 215) (current study)	77.2%	12.1%	10.7%	Tube agglutination

**Table 3 animals-10-01129-t003:** Alloantibody prevalence of AB blood group system in cats from regions in Northern (Lombardy) and Southern (Sicily) Italy.

		Whole Population (*n* = 327)	Origin		*p* Value
			Northern Italy (*n* = 196)	Southern Italy (*n* = 131)	
**Alloantibodies**	**Presence**	62 (19.0%)	30 (15.3%)	32 (24.4%)	0.7015
	**Absence**	265 (81.0%)	166 (84.7%)	99 (75.6%)	
**Alloantibodies by blood type**	**A**	30/268 (11.2%)	19/177 (10.7%)	11/91 (12.1%)	0.7398
	**B**	32/34 (94.1%)	11/12 (91.7%)	21/22 (95.5%)	0.6585
	**AB**	0/25 (0.0%)	0/7 (0.0%)	0/18 (0.0%)	-

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
