# Peer review of "Prevalence of Blood Types and Alloantibodies of the AB Blood Group System in Non-Pedigree Cats from Northern (Lombardy) and Southern (Sicily) Italy"

_animals, 2020, doi:10.3390/ani10071129_

Round 1
Reviewer 1 Report
Dear Authors,
This is an interesting study. I have made several comments about specific points of the script and then some more general points at the end. I hope they are useful.
Sentences beginning line 52: It is probably better to use the term haemolytic transfusion reaction instead of major and minor transfusion reaction. So perhaps: ‘A marked acute haemolytic transfusion reaction generally occurs [5] (which may be fatal) if the type-B recipient receives type-A or AB blood. Less severe acute haemolytic transfusion reactions (mTR) lead to the premature destruction of transfused red cells in type-A cats receiving type-B or AB blood. [6]
Line 54: The way this is worded it sounds as if the AB recipient destroys the transfused cells, whereas the haemolytic reaction occurs because of anti-A or anti-B alloantibodies in the donor plasma.
Line 58: Suggest change ‘as anti-A alloantibodies recognize AB RBCs’ to ‘as AB RBCs also have A antigen on their cell surface’
Line 91: the exclusion criteria is a ‘lack’ of owner consent
Line 92: The sample size calculation needs to describe what the aim of the study was to make sense. So, why did the estimated prevalance of AB cats matter? Was it because you wanted to see if there was a difference in prevalence of AB cats in North and South Italy? If so you need to give an idea of what you thought the difference was going to be and then the power you aimed for.
Line 96: How was province determined in the Northen Italy population?
Line 103: Suggest removal of sentence ‘We evaluated some data on TUBE that were not available in literature.’ A brief overview of the TUBE technique is required here and a reference to show that it is a recognised technique.
Line 113: Suggest a brief overview of the technique is provided
What is the origin of the colony cats? Are they related cats bred for purpose? What are the colonies used for?
Line 228: You have not calculated the risk of MTR if a type B cat is given type A or AB blood, you have calculated the risk of a B cat being administered type A or AB blood if the type of neither recipient or donor is known which is subtly but importantly different. To the reader, the way the result is currently written, it sounds ilke it’s only a 6-10% chance of having a MTR if you have a type B cat and you don’t bother to type the donor which is not true. Also, given you know the alloantibody rate here, you haven’t taken this into account, so actually you probably can’t say the risk of reaction, probably more that you can say the risk of mis-matched transfusion.
Sentence beginning Line 263: The prevalence of type B in the {12} study is within your CIs for all cats, so either you need CIs for just the non-pedigree cats which show this number is outside it, or you cannot postulate about changes over time as it may be there is no change. Also, it could be due to differences in sampling and typing methods between the studies, there are other possible explanations.
Line 281: This sentence seems to suggest that type AB cats may be more resistant to FeLV which is a large jump to make. I don’t think this is what you are meaning,, but it is not clear. I would suggest removing this sentence.
Could it just be that the sampled population is different between N and S Italy, more detail is required about the colony cats.
Line 312: This should be reported in results not discussion
Line 336: It shows that NI is ‘likely’ a common event, the study doesn’t actually see whether NI actually occurs.
Line 348: Alloantibody levels were not ‘always’ determined.
Further questions:
- Probably the most important point to address is the difference in populations between the north and South Italy samples, can they really be said to be equivalent representations of their locality?
- Did cats travel to north Italy hospital?
- The fact that cats were presenting to a hospital is a confounding factor, as owners may be more likely to present cats that are valuable, perhaps why there are more pure breeds. But also, it could be that blood type is asscoaited with susceptibilyt to certain diseases and so may mean that blood type presentation is not represtntative of the local population.
- The testing of the reproducibility of the TUBE method wasn’t asscoaited with the aims and seems like a separate project.
- No back typing of type A cats
- There is a massive oversimplification with percent calculations assuming that that recipient cats come from the same population as donors and also that matings are randomly from these populations as well which is very unlikely.
Author Response
Dear Authors,
This is an interesting study. I have made several comments about specific points of the script and then some more general points at the end. I hope they are useful.
We would like to thank the reviewer for the time and effort taken to review this manuscript and very useful comments
Sentences beginning line 52: It is probably better to use the term haemolytic transfusion reaction instead of major and minor transfusion reaction. So perhaps: ‘A marked acute haemolytic transfusion reaction generally occurs [5] (which may be fatal) if the type-B recipient receives type-A or AB blood. Less severe acute haemolytic transfusion reactions (mTR) lead to the premature destruction of transfused red cells in type-A cats receiving type-B or AB blood. [6]
R: thank you for the suggestion, we have changed the sentence accordingly.
Line 54: The way this is worded it sounds as if the AB recipient destroys the transfused cells, whereas the haemolytic reaction occurs because of anti-A or anti-B alloantibodies in the donor plasma.
R: we have corrected as follow at line 56-59:” Less severe acute haemolytic transfusion reactions lead to the premature destruction of transfused red cells in type-A cats receiving type-B or AB blood or type-AB cats receiving type-B and A blood, due to anti-A or anti-B alloantibodies in the donor plasma [6].”
Line 58: Suggest change ‘as anti-A alloantibodies recognize AB RBCs’ to ‘as AB RBCs also have A antigen on their cell surface’
R: done
Line 91: the exclusion criteria is a ‘lack’ of owner consent
R: following the suggestion of all reviewers we have modified and corrected this paragraph as follows at lines 90-93: ”Ethylenediamine tetra-acetic acid (EDTA)-anticoagulated surplus feline blood samples collected for various clinical reasons at the University of Milan (Northern Italy) and at the University of Messina (Southern Italy) were randomly included in the study between November 2014 and December 2015.”
Line 92: The sample size calculation needs to describe what the aim of the study was to make sense. So, why did the estimated prevalance of AB cats matter? Was it because you wanted to see if there was a difference in prevalence of AB cats in North and South Italy? If so you need to give an idea of what you thought the difference was going to be and then the power you aimed for.
R: you are right, we have added more justification of the sample size calculation to the hypothesis of the study in the Introduction section as follows at lines 80-85: “As type-A blood is the most prevalent in all non-pedigree cats worldwide, the hypothesis of the study was that cats from Sicily may have a different prevalence of the most rare blood types-B and type-AB and a different prevalence of alloantibodies to northern cats. The rationale being that these cats were from an island where a small genetic pool combined with inbreeding could have increased the number of cats with rare blood types over time.”
Line 96: How was province determined in the Northen Italy population?
R: with the postcode as for the southern cats, we added this information as follows at line 114: “Postcode was used to determine the province of origin for cats.
Line 103: Suggest removal of sentence ‘We evaluated some data on TUBE that were not available in literature.’ A brief overview of the TUBE technique is required here and a reference to show that it is a recognised technique.
R: we removed the sentence and have given brief overview of the TUBE technique and we added references to show that it is a recognised technique. You can find this information as follows at lines 117-139: “Blood typing was performed using the tube agglutination technique (TUBE) as previously described[23,35,36]. Briefly, to prepare a RBC suspension, 0.5 mL of EDTA-blood was centrifuged at 1,600 g for 10 minutes at room temperature. Plasma was collected into a separate tube and was stored at –20°C until tested for alloantibodies screening. The RBC pellet was washed by adding 2.5 mL of isotonic 0.9% NaCl solution. Following centrifugation at 1,000 g for 1 minute, the supernatant was removed, and the pellet was resuspended and then recentrifuged twice and finally reconstituted to a 5% RBC suspension. Polyclonal antibodies contained in type-B cat plasma (obtained from a type-B blood donor, collected with CPD anticoagulant at 1:7 ratio, stored at -20°C) were used as primary reagents for the detection of type A red cell antigens. Trititum vulgaris lectin (Sigma-Aldrich, 8 μg/mL) was used for the detection of type-B RBCs antigens as this lectin binds to the NeuAc terminal of the type-B ganglioside and therefore strongly agglutinates feline type-B RBCs, but does not agglutinate type-A RBCs [35]. A 0.9% NaCl solution (saline solution) was used as a negative control. In 3 glass tubes (PYREX® Tube Borosilicate glass 12x75mm, Coming Inc., NewYork USA), 50 μL of 5% RBC suspension was mixed with 100 μL of type-B plasma (anti-A reagent, tube A), 100 μL of Triticum vulgaris lectin solution (anti-B reagent, tube B), or 100 μL of saline (control reagent, tube C), respectively. These mixtures were incubated at room temperature for 15 minutes before centrifugation for 15 seconds at 1,000 g. Tubes were then gently agitated, and blood type was recorded for the tube where macroscopic agglutination was present (Figure 1).Type-B and AB samples were confirmed by back typing technique as previously described [12,37].
As samples collected in Sicily were tested after a minimum of 10 days from sampling, we preliminarily assessed the stability of samples by typing one sample of each blood type, stored at 4±2°C, using TUBE after 24, 48, 72, and 96 hours, and after 1, 2, 3, and 4 weeks of storage. Each sample stored at 4±2°C for up 1 month was correctly blood typed by TUBE.”
Line 113: Suggest a brief overview of the technique is provided
R: we provided a brief overview of the alloantibody screening as follow at lines 147-159: ”Alloantibody screening is a modified major crossmatch, using RBCs of a known blood group, thus screening for naturally alloantibodies against that group in the recipient. Alloantibody screening was performed as previously described on frozen plasma samples [37–39]. Briefly, in a glass tube25 μL of the type-A RBC suspension (5% in 0.9% NaCl) was added to 50 μL of type-B plasma. The same was done with type-B RBCs and type-A plasma and with type-A and type-B RBCs with type-AB plasma. The solution was gently mixed, incubated for 15 minutes at room temperature and then centrifuged for 15 seconds at 1,000 g. Agglutination was evaluated by gently agitating the tube to resuspend the non-agglutinating RBCs in the cell button. The presence of alloantibodies against RBCs of a different blood type was identified if macroscopic agglutination was present, as in the previous description of the tube technique for blood typing. In addition, where agglutination was not macroscopically visible, microscopic agglutination was evaluated at x40. Any RBC agglutination, macroscopic and/or microscopic, but not rouleaux presence alone, was considered a positive result for presence of alloantibodies in the tested plasma sample.”
What is the origin of the colony cats? Are they related cats bred for purpose? What are the colonies used for?
R: colony cats are stray cats which live in colonies. More information on colony stray cats has been provided in the Material and Methods section as follows at lines 93-105: “Samples from all types of Italian cats were included in this study, ie samples collected from: I) owned cats presented for routine check ups before neutering, or for diagnostic purposes in the case of unhealthy cats, II) shelter cats before surgery for neutering, III) stray colony cats. This last group comprised stray cats that live in colonies in urban and rural areas of Italian cities and are captured for a trap, neuter, and release (TNR) sterilization program as part of a national program to control stray pet populations under Italian National Law (law no. 281/1991).”
Line 228: You have not calculated the risk of MTR if a type B cat is given type A or AB blood, you have calculated the risk of a B cat being administered type A or AB blood if the type of neither recipient or donor is known which is subtly but importantly different. To the reader, the way the result is currently written, it sounds ilke it’s only a 6-10% chance of having a MTR if you have a type B cat and you don’t bother to type the donor which is not true. Also, given you know the alloantibody rate here, you haven’t taken this into account, so actually you probably can’t say the risk of reaction, probably more that you can say the risk of mis-matched transfusion.
R: as suggested, we have deleted all the calculations about major and minor transfusion reactions as these calculations were very theoretic since they assume that that recipient cats come from the same population as donors which is very unlikely.
Sentence beginning Line 263: The prevalence of type B in the {12} study is within your CIs for all cats, so either you need CIs for just the non-pedigree cats which show this number is outside it, or you cannot postulate about changes over time as it may be there is no change. Also, it could be due to differences in sampling and typing methods between the studies, there are other possible explanations.
R: you are right, we removed this paragraph as it seemed too speculative
Line 281: This sentence seems to suggest that type AB cats may be more resistant to FeLV which is a large jump to make. I don’t think this is what you are meaning,, but it is not clear. I would suggest removing this sentence.
R: we have removed the sentence as suggested
Could it just be that the sampled population is different between N and S Italy, more detail is required about the colony cats.
R: you can find more details on colony stray cats at lines 93-105 as follows: “Samples from all types of Italian cats were included in this study, ie samples collected from: I) owned cats presented for routine check ups before neutering, or for diagnostic purposes in the case of unhealthy cats, II) shelter cats before surgery for neutering, III) stray colony cats. This last group comprised stray cats that live in colonies in urban and rural areas of Italian cities and are captured for a trap, neuter, and release (TNR) sterilization program as part of a national program to control stray pet populations under Italian National Law (law no. 281/1991).”
Line 312: This should be reported in results not discussion
R: we moved this information to the results section, obviously changing the data with the new corrected data based only on non-pedigree DSH cats evaluation
Line 336: It shows that NI is ‘likely’ a common event, the study doesn’t actually see whether NI actually occurs.
R: as suggested, we have deleted all the calculations on neonatal isoerythrolysis as these calculations were very theoretic - assuming that matings are random from these populations which is very unlikely. In addition the evaluated risk of NI does not take into account that, for a variety of reasons, some kittens do not absorb any colostral antibodies and therefore cannot develop NI. Finally, the population typed in the current study consisted of non-pedigree DSH, which are generally not purpose-bred, therefore the calculated risk of NI remain a theoretical risk. Therefore we decided to delete all data on potential risk of neonatal erythrolysis in our populations.
Line 348: Alloantibody levels were not ‘always’ determined.
R: corrected as suggested
Further questions:
Probably the most important point to address is the difference in populations between the north and South Italy samples, can they really be said to be equivalent representations of their locality?
R: we have better explained the sample cat population used in the study and that not all sampled cats were sick cats. I hope it is now clear that we tried to sample as representative a feline population as possible.
Did cats travel to north Italy hospital?
R: no it was the samples that travelled to a northern Italy laboratory to be analysed. We added more information on this as follows at lines 106-109: ”All analyses were performed at the Veterinary Transfusion Research Laboratory (REVLab) of the University of Milan where the Northern Italy cats were sampled. Blood samples from cats in Southern Italy were stored at 4-6°C after collection and were shipped refrigerated to the REVLab where they were analyzed within 48 hours of arrival.”
The fact that cats were presenting to a hospital is a confounding factor, as owners may be more likely to present cats that are valuable, perhaps why there are more pure breeds. But also, it could be that blood type is asscoaited with susceptibilyt to certain diseases and so may mean that blood type presentation is not represtntative of the local population.
R: we have better explained that the samples from the feline sample population used in the study come from both healthy and unhealthy cats. We hope that it is now more clear that we tried to sample as representative a feline population as possible.
The testing of the reproducibility of the TUBE method wasn’t asscoaited with the aims and seems like a separate project.
R: you are right and we deleted it
No back typing of type A cats
R: usually only type B and AB blood samples undergo back typing as you can reverse type these samples to find alloantibodies against type-A RBCs (in case of type-B samples) or find no alloantibodies (in case of type-AB samples). This is not possible with type-A samples as no all type-A cats have alloantibodies against type-B RBCs.
There is a massive oversimplification with percent calculations assuming that that recipient cats come from the same population as donors and also that matings are randomly from these populations as well which is very unlikely.
R: you are right, therefore we decided to delete all the calculations about major and minor transfusion reactions and neonatal isoerythrolysis as these calculations were very theoretical. In addition, the evaluated risk of NI does not take into account that for a variety of reasons, some kittens do not absorb any colostral antibodies and therefore cannot develop NI. In addition, the population typed in the current study comprised non-pedigree DSHs, which are generally not purpose-breed, therefore the calculated risk of NI remained a theoretical risk.
Reviewer 2 Report
General Comments:
This study provides additional information concerning blood types in the AB blood group system in cats in Italy, as well as information concerning the presence of alloantibodies in type-A and type-B cats. The relatively high prevalence of type-AB in non-pedigree cats in southern Italy is noteworthy. The inclusion of various purebred cats, especially since there are many more in the northern group, makes the data more difficult to interpret. It is further complicated by differing numbers of various breeds, especially since Ragdoll cats have been reported to differ considerably in blood types compared to DSH cats. I recommend only using data from non-pedigree DSH cats in this manuscript. This will shorten the paper and make it easier to understand. Aside from Ragdoll cats, there are not enough of any other breed to report on, and even the Ragdolls might be related, which could be give misleading results.
Specific Comments and Questions
- Line 28. The use of abbreviations nI and sI are not needed, because they are not used in the text and nI might be confused with NI, which is used in the abstract and text.
- Abstract. Does not indicate this is a mix of pedigree and non-pedigree cats. The whole paper as well as abstract should focus on non-pedigree cats.
- Line 48. Statement says all type-B cats have alloantibodies against type-A erythrocytes, however, this is not consistent with results in this paper. Suggest rewording like ‘All type B cats have been reported to have alloantibodies against type-A erythrocytes.
- Line 49. Reference 3 indicates approximately 1/3 of type-A cats have macroscopic agglutination, but these authors found 30 of 30 cats without macroscopic agglutination had microscopic agglutination and positive Coombs tests. This suggests that most type-A cats have natural antibodies that can be identified when more sensitive tests are done.
- Line 55. Type B should be type-B for consistency. Elsewhere in text a hyphen is sometimes not used between type and A, B, or AB (example line 65)
- Line 60. This paragraph needs some rewording. Suggest starting with “Type-A is the most prevalent blood type worldwide. Type-B and type-AB blood types were reported to be rare in initial studies of nonpedigree cats. However …
- Lines 87 and throughout methods and results. All italic subheadings begin with 3.3. Is this to indicate subheadings in italics or should numbers be different?
- Line 90. I am not sure what prospectively randomly sampled means when using surplus blood.
- Line 96. What are “colony cats”. How are they different from owned and shelter cats? Are these related cats in a breeding colony that might alter the frequency of blood types?
- Lines 103-108. This paragraph fits better after the following paragraph.
- Lines 103-104. Sentence needs rewording.
- Line 105. Reword to make clearer ‘’by running one sample each of type’..
- Line 148. Change “multiple times” to “10 times each” to be more specific.
- Table 1 is not needed if only DSH cats are compared. Statements concerning sex and origin can be given in the text in the methods section.
- Table 2 and Figure 1 are redundant so the one of these is not needed. This data should be recalculated and presented with no purebred cats included.
- Lines 181-183. Data for non-pedigree DSH cats only should be used to replace mixed data in Table 2. Differences are more dramatic and this removes questions concerning contributions of purebred cats.
- Lines 195 and 196. Statistics are given, but no data is given concerning purebred cats. Again, adding purebred cats to the mix adds confusion. We already know certain that certain breeds have higher type-B and type AB blood than DSH cats.
- Table 3. This table could replace table 1.
- Table 4. Redo with only DSH cats.
- Lines 228 – 235. Redo risk calculations with only non-pedigree cats. These values would not be valid if one is going to give a transfusion to some purebred cats where type-B prevalence is reported to be high.
- The discussion should be shortened and focus on major differences. There seems to be an overemphasis in detailed comparisons to other countries or parts of countries. This is the type of information that might be included in a large table in a review article.
- Line 264. Should indicate “southern Italy.”
- Line 266-367. Reword sentence for clarity.
- Line 269. How many cats were tested in central Italy? If only a few, results may not reflect reality.
- Lines 277-279. A small gene pool seems more likely than genetic drift. If there is a selective advantage for type AB, one might expect it to be much higher in other areas. Genetic drift seems to be considered for evolutionary changes. Does it occur rapidly, as might occur in a few years in cats?
- Lines 279-281. Highly speculative. It is hard to understand how a change in blood phenotype would be linked to immunity or reproduction.
- Lines 305-316. Tests for alloantibodies appear to be more subjective than blood typing, especially in the interpretation of microscopic agglutination. Consequently, testing methods and differences in individuals to accurately recognize microscopic agglutination may be more important than geographic locations. In addition, this sentence is redundant to lines 308-310.
Author Response
General Comments:
This study provides additional information concerning blood types in the AB blood group system in cats in Italy, as well as information concerning the presence of alloantibodies in type- A and type-B cats. The relatively high prevalence of type-AB in non-pedigree cats in southern Italy is noteworthy. The inclusion of various purebred cats, especially since there are many more in the northern group, makes the data more difficult to interpret. It is further complicated by differing numbers of various breeds, especially since Ragdoll cats have been reported to differ considerably in blood types compared to DSH cats. I recommend only using data from non-pedigree DSH cats in this manuscript. This will shorten the paper and make it easier to understand. Aside from Ragdoll cats, there are not enough of any other breed to report on, and even the Ragdolls might be related, which could be give misleading results.
R: We would like to thank the reviewer for the time and effort taken to review this manuscript and very useful comments
Thank you for your suggestion. In the revised version of the manuscript we used only data from non-pedigree DSH cats, therefore we didn’t include purebred cats and samples from cats with no reported breed
Specific Comments and Questions
- Line 28. The use of abbreviations nI and sI are not needed, because they are not used in the text and nI might be confused with NI, which is used in the abstract and text. 

R: we deleted the abbreviation nI and Si
2. Does not indicate this is a mix of pedigree and non-pedigree cats. The whole paper as well as abstract should focus on non-pedigree cats. 

R: now the manuscript focused only on non-pedigree DSH cats
- Line 48. Statement says all type-B cats have alloantibodies against type-A erythrocytes, however, this is not consistent with results in this paper. Suggest rewording like ‘All type B cats have been reported to have alloantibodies against type-A erythrocytes. 

R: you are right, we changed as suggested
- Line 49. Reference 3 indicates approximately 1/3 of type-A cats have macroscopic agglutination, but these authors found 30 of 30 cats without macroscopic agglutination had microscopic agglutination and positive Coombs tests. This suggests that most type- A cats have natural antibodies that can be identified when more sensitive tests are done. 

R: we added this information in the Introduction as follow at lines 50-52: “All type-B cats have been reported to have alloantibodies against type-A erythrocytes, approximately one third of type-A cats have low-titer macroscopic anti-B agglutinating alloantibodies, but no alloantibodies have been found in type-AB cats [3].”
In the Discussion section as follow at lines 278-283: ”Other than geographical variations, testing methods and differences in individuals to accurately recognize microscopic agglutination may be influence the results of alloantibodies screening in previous studies. In addition, it is possible that the delay in separating plasma from RBCs in some samples, and the long storage time before analysis, could have influenced the results of alloantibody screening in our study, reducing, and thus limiting, recognition of the weak RBC agglutination reactions characteristic of type-A alloantibodies.”
- Line 55. Type B should be type-B for consistency. Elsewhere in text a hyphen is sometimes not used between type and A, B, or AB (example line 65) 

R: done
- Line 60. This paragraph needs some rewording. Suggest starting with “Type-A is the most prevalent blood type worldwide. Type-B and type-AB blood types were reported to be rare in initial studies of nonpedigree cats. However ... 

R: done as suggested
7. Lines 87 and throughout methods and results. All italic subheadings begin with 3.3. Is this to indicate subheadings in italics or should numbers be different? 

R: it is an error and all numbers in italic subheading have been deleted
- Line 90. I am not sure what prospectively randomly sampled means when using surplus blood. 

R: we corrected with “randomly included”. The sentence is as follows at lines 90-93: “Ethylenediamine tetra-acetic acid (EDTA)-anticoagulated surplus feline blood samples collected for various clinical reasons at the University of Milan (Northern Italy) and at the University of Messina (Southern Italy) were randomly included in the study between November 2014 and December 2015.”
- Line 96. What are “colony cats”. How are they different from owned and shelter cats? Are these related cats in a breeding colony that might alter the frequency of blood types? 

R: they are stray cays that in Italy live in colonies. More information on colony stray cats has been added in the M&M section as follows at lines 93-99: “Samples from all types of Italian cats were included in this study, ie samples collected from: I) owned cats presented for routine check ups before neutering, or for diagnostic purposes in the case of unhealthy cats, II) shelter cats before surgery for neutering, III) stray colony cats. This last group comprised stray cats that live in colonies in urban and rural areas of Italian cities and are captured for a trap, neuter, and release (TNR) sterilization program as part of a national program to control stray pet populations under Italian National Law (law no. 281/1991).”
- Lines 103-108. This paragraph fits better after the following paragraph. 

R: moved after the tube technique description as suggested
- Lines 103-104. Sentence needs rewording. 

R:as suggested by another reviewer this sentence has been removed.
- Line 105. Reword to make clearer ‘’by running one sample each of type’.. 

R: as suggested by another reviewer we removed the reproducibility evaluation of the Tube method as it is outside the aim of this study
13.Line 148. Change “multiple times” to “10 times each” to be more specific.
R: as suggested by another reviewer we removed the reproducibility evaluation of the tube method as it is outside the aim of this study
- Table 1 is not needed if only DSH cats are compared. Statements concerning sex and 
origin can be given in the text in the methods section. 

R: as suggested we moved all info from table 1 to the Results section at line 176-184 as follow: “Age was not recorded in 42 cats (36 from Northern Italy, 6 from Southern Italy). Median age in Northern Italian cats (median 3 yrs, range 0.3-18 yrs) was significantly higher than the Southern ones (median 2 yrs, range 0.3-19 yrs) (P<0.0001). Sex was not reported in 20 cats (all from Northern Italy) and no significant difference in male and female prevalence was found between northern (male n=102, female n=111) and southern (male n=96, female n=119) cats (P=0.5025). A significantly higher number of owned cats were included in the northern (n=120) population than from Southern Italy (n=64) (P<0.0001) and a higher number of colony stray cats were from Southern (n=115) rather than from Northern Italy (n=79) (P<0.0001).”
- Table 2 and Figure 1 are redundant so the one of these is not needed. This data should 
be recalculated and presented with no purebred cats included. 

R: we left the Table (that is now table 1) and deleted Figure 1. Now you can find only data on non-pedigree DSH cats
- Lines 181-183. Data for non-pedigree DSH cats only should be used to replace mixed 
data in Table 2. Differences are more dramatic and this removes questions concerning 
contributions of purebred cats. 

R: we deleted all the data relating to purebred cats and cats with no reported breed.Now you can find only data on non-pedigree DSH cats
- Lines 195 and 196. Statistics are given, but no data is given concerning purebred cats.
Again, adding purebred cats to the mix adds confusion. We already know certain that certain breeds have higher type-B and type AB blood than DSH cats.
R: we deleted all the data relating to purebred cats and cats with no reported breed. Now you can find only data on non-pedigree DSH cats
- Table 3. This table could replace table 1. 

R: table 3 is now table 2
- Table 4. Redo with only DSH cats. 

R: done only with non-pedigree DSH cats. This is now table 3
- Lines 228 – 235. Redo risk calculations with only non-pedigree cats. These values would not be valid if one is going to give a transfusion to some purebred cats where type-B prevalence is reported to be high.

R: as suggested by another reviewer, we have deleted all the calculations about major and minor transfusion reactions and neonatal isoerythrolysis as these calculations were very theoretical - assuming that that recipient cats come from the same population as donors and also that matings are random from these populations which is very unlikely. In addition the evaluated risk of NI does not take into account that for a variety of reasons, some kittens do not absorb any colostral antibodies and therefore cannot develop NI. In addition, the population typed in the current study consisted of non-pedigree DSH, which are generally not purpose-bred, therefore the calculated risk of NI remains a theoretical risk.
- The discussion should be shortened and focus on major differences. There seems to be an overemphasis in detailed comparisons to other countries or parts of countries. This is the type of information that might be included in a large table in a review article.
R: we tried to shorten the discussion and focus it on major differences.
22.Line 264. Should indicate “southern Italy.”
R: done
- Line 266-367. Reword sentence for clarity. 

R: reworded as follows at lines 244-246: “Prevalence of type-AB blood in previous Italian studies varied in non-pedigree cats from 2.1% [12]to 5.7% [32]in Northern Italy, with the highest prevalence (18%) found only in pedigree Italian Ragdoll cats in catteries located in Northern Italy [37].”
- Line 269. How many cats were tested in central Italy? If only a few, results may not reflect reality. 

R: the study tested 401 cats therefore we decided to maintain it
- Lines 277-279. A small gene pool seems more likely than genetic drift. If there is a 
selective advantage for type AB, one might expect it to be much higher in other areas. Genetic drift seems to be considered for evolutionary changes. Does it occur rapidly, as might occur in a few years in cats? 

R: you are right, a small genetic pool combined with inbreeding could be the more probable explanation for the unexpectedly high proportion of type-AB cats. We deleted previous comments on genetic drift. Now you can find this paragraph as follows at line 253-258: “The high prevalence of type-AB cats in the Southern Italy cat population may reflect the relatively small gene pool from which this population has developed. Sicily is separated from the mainland by a marine natural barrier that prevents free movement of terrestrial fauna and the non-pedigree cat population is isolated from neighboring areas. These circumstances may have enhanced inbreeding in Sicilian domestic cats as is recognized in wild cats [50].”
26.Lines 279-281. Highly speculative. It is hard to understand how a change in blood phenotype would be linked to immunity or reproduction.
R: you are right, we deleted this sentence
27.Lines 305-316. Tests for alloantibodies appear to be more subjective than blood typing, especially in the interpretation of microscopic agglutination. Consequently, testing methods and differences in individuals to accurately recognize microscopic agglutination may be more important than geographic locations. In addition, this sentence is redundant to lines 308-310.
R: we added some limitations relative to our study on alloantibodies as follow at lines. The new paragraph is as follows at lines 278-283:” Other than geographical variations, testing methods and differences in individuals to accurately recognize microscopic agglutination may be influence the results of alloantibodies screening in previous studies. In addition, it is possible that the delay in separating plasma from RBCs in some samples, and the long storage time before analysis, could have influenced the results of alloantibody screening in our study, reducing, and thus limiting, recognition of the weak RBC agglutination reactions characteristic of type-A alloantibodies.”
Reviewer 3 Report
Dear Authors,
thank you for your interesting manuscript about an epidemiological study of blood groups and alloantibodies in cats from 2 different regions of Italy. The data shown are useful to deepen the knowledge relating to the distribution of blood groups and the frequency of alloantibodies in cats in Italy, given the limited availability of data on this matter. This manuscript has, therefore, a mostly regional relevance within a nation (Italy) and could be worthy of publication after the elucidation and revision of some important inclusive criteria and methodological implications. The language of the manuscript should be revised.
General comments
- The title is not appropriate since data mainly come from DSH cats (there are very few purebred cats and it is preferable to remove them since they do not give additional information. Moreover, results of the prevalence of blood groups in the different breeds are not shown in Table 2) and samples come from only 2 Italian regions (sample from Calebria are too few to include this region).
A proposal can be: Prevalence of blood types and alloantibodies of the AB blood group system in DSH cats from Northern Italy (Lumbardy) and Southern Italy (Sicily)
-- Section of Material and Methods must be improved. Particularly, sample inclusion criteria and analysis, as well as the technique used for blood typing (TUBE) has to be well described. - Statistical analysis should be calculated excluding purebred cats since they are very few and almost all from northern Italy. Furthermore, the number of samples from the 2 regions of southern Italy should be better specified. I can understand that only 16 samples from Calebria have been analyzed. They should be removed because not indicative of the Region.
-- Section of results should be changed by removing the results from purebred cats since they are known to have different blood group frequencies and can alter results. In this study, the presence of a few purebred cats does not give us more information (indeed, probably removing them the statistical differences between northern and southern samples could increase). Moreover, they are almost all from northern Italy. In table 2 it is not specified how many blood groups were obtained from each breed.
The group without breed specification has been added in DSH group? Please clarify and think to exclude these data.
Specific Comments:
Summary
Line 24: change “useful” with “ usefulness”
Abstract:
Change the total number of samples and prevalences excluding purebred cats and cats from Calebria Region. Cats without breed specification? Should be excluded or separated.
Introduction
Line 44 The most clinically important
Lines 46-47 It is not clear, please change
Line 56 please change with type A and AB kittens and delate from “Kittens …..to …..RBCs”
Line 77 I suggest deleting Calebria Region, since the sample number from this region probably (you have not reported the number but t is possible to extrapolate it) is very low (15-16 samples)
Line 80 please, change with …from the Southern region (Sicily)….compared to cats from the northern region (Lumbardia region)
Material and Methods
Line 87: 3.3 is probably a repeated error
Samples and population
Lines 91-92 : the inclusion of potentially sick cats could affect the results of the study. Results of blood typing can be affected by disease states of the animals. Particularly FeLV, anemia, and autoagglutination may contribute to the test inaccuracies as shown by Seth et al. 2011. Could you describe the diseases affecting those sick cats included in the study? Could you include the number of sick cats and the type of disease (especially if FeLV-anemia and autoagglutinations occurred) in the results section and Table1? Could you include in the discussion this possible limitation? Did you perform FeLV test in all cats?
Also, please, specify how long the samples have undergone to storage. Were the blood groups carried out immediately after collection with the tube test or stored? Since the analysis was conducted in northern Italy, the samples sent from southern Italy were refrigerated? Has it been used Frozen plasma?). The storage time and method used may have affected some results of blood typing due to stability related problems of samples during the time
Line 98: delete the phrase “This study…..only”.
TUBE test
Line 103-108: please rewrite and describe deeply the TUBE method used with reagents and all the methodological steps. Do you have any image of the results?
Probably you decided to use the Tube agglutination test since it is considered the original gold standard method for blood typing and alloantibody screening based on 100% sensitivity and specificity for detection of the A antigen. Nevertheless, this method has been demonstrated to have an excellent concordance of typing results by the use of other more rapid and objective techniques using monoclonal antibodies. Did you use the rapid and accurate immunochromatographic test kits commercially available to compare and confirm blood groups?
Tube test could induce errors and inaccuracies in case of autoagglutination, severe anemia, and or FeLV+ cats since the interpretation of results is subjective and sometimes the presence of alloantibodies could be strongly reduced in these sick cats. I
In some of these cases, the immunochromatographic test kits could be more objective to detect blood group, especially in the case of diseased cats.
What do you mean for “we evaluated some data….literature”? Please rewrite.
Line 104: please, write “and” a microscopic evaluation is necessary
LINE 131 “as per” is an error? Please, add citations
Line 153 how many samples were analyzed from Sicily and how many from Calebria? Unclear data. It is possible to extrapolate a very low number of samples from Calabria (15-16) and I suggest to remove them.
LINES 157-161: purebred cats (56 and the majority from Northern Italy) and cats with no breed reported (41) should be excluded since each breed tends to have a specific prevalence of blood groups and this could distort the epidemiological data. Moreover, each breed has a low number of samples and the Ragdoll cats (the pure breed mostly represented, only from Northern Italy), are potentially AB and B types.
If you want to include these cats, please add in table 2 the results of blood typing dividing into 3 groups: each breed, cats with no breed data, and DSH. It allows to better evaluate the distribution of blood groups
In the text (153-166) and in table 1 is not clearly reported the number of DSH cats and purebred cats
In table 1 cats with no breed, data are included in DSH?
LINE 176-185 : I suggest to recalculate the percentage excluding purebred cats, cats from Calebria and cats with no breed data
LINE 194-196: delete. Obviously data are not valid due to the low number of samples from purebred cats, (the majority from Northern Italy).
LINES 197-200 and table 3: please, delete data from Calebria region since the number of samples are too low
LINE 214: please, specify how many type A, type B, and type AB cats have been tested for alloantibodies.
DISCUSSION
Please, reshape the discussion by removing at least the data from purebred cats and those from Calebria. These groups are too small and do not give useful valuable informations for discussion.
When you write “Southern and Northern Italy”, I suggest specifying that they are regions.
Line 263: this sentence is questionable. The number of individual groups (DSH, purebred cats) from the individual regions is too low and the sample is not representative of the entire population of cats in the considered areas. Furthermore, it is possible that the use of different blood typing kits or tests applied in the previous studies could induce differences in the interpretation and comparison of results.
Lines 264-265: Are the percentage you described obtained excluding purebred cats? Please, clarify.
Lines 276-279: Could you better explain this concept? Please, add citations and clarify
Lines 286-287: from “where….. to…. 9.7%” please, delate.
Lines 287-290: this statement can’t be demonstrated. Cats with AB blood group are DHS cats? Purebreed cats?
Line 300: What do you mean with the sentence “the first time stability of samples”?
LINES 300-304: Did you stored blood EDTA samples for 1 month before doing the analysis? It is not explained in the material and methods sections. Is the stability of samples maintained? Could you add some bibliography concerning this method and the stability of samples (RBCs antigens and alloantibodies)
Line 320: 12 weeks or 3 months of age?
Lines 347-349: it is not specified the number of A, B, and AB cats in the purebred cats.
Lines 348-349: “Due to the limited….determined” What samples do you mean?
Conclusions
Line 360: please, change with “a Region of Southern Italy….had a higher……and a lower prevalence…..compared to a Region of Northern Italy”
Author Response
Dear Authors,
thank you for your interesting manuscript about an epidemiological study of blood groups and alloantibodies in cats from 2 different regions of Italy. The data shown are useful to deepen the knowledge relating to the distribution of blood groups and the frequency of alloantibodies in cats in Italy, given the limited availability of data on this matter. This manuscript has, therefore, a mostly regional relevance within a nation (Italy) and could be worthy of publication after the elucidation and revision of some important inclusive criteria and methodological implications. The language of the manuscript should be revised.
We would like to thank the reviewer for the time and effort taken to review this manuscript and very useful comments
General comments
- The title is not appropriate since data mainly come from DSH cats (there are very few purebred cats and it is preferable to remove them since they do not give additional information. Moreover, results of the prevalence of blood groups in the different breeds are not shown in Table 2) and samples come from only 2 Italian regions (sample from Calebria are too few to include this region).
A proposal can be: Prevalence of blood types and alloantibodies of the AB blood group system in DSH cats from Northern Italy (Lumbardy) and Southern Italy (Sicily)
R: thank you for your suggestion, we have changed the title as suggested
Section of Material and Methods must be improved. Particularly, sample inclusion criteria and analysis, as well as the technique used for blood typing (TUBE) has to be well described. - Statistical analysis should be calculated excluding purebred cats since they are very few and almost all from northern Italy. Furthermore, the number of samples from the 2 regions of southern Italy should be better specified. I can understand that only 16 samples from Calebria have been analyzed. They should be removed because not indicative of the Region.
R: we have improved sample inclusion criteria and better described the feline population studied in the Materials and Methods section as follows at lines 93-105: “Samples from all types of Italian cats were included in this study, ie samples collected from: I) owned cats presented for routine check ups before neutering, or for diagnostic purposes in the case of unhealthy cats, II) shelter cats before surgery for neutering, III) stray colony cats. This last group comprised stray cats that live in colonies in urban and rural areas of Italian cities and are captured for a trap, neuter, and release (TNR) sterilization program as part of a national program to control stray pet populations under Italian National Law (law no. 281/1991).”
The technique used for blood typing (TUBE) has been better described as well as alloantibodies screening. You can find this new information in the M&M section as follows at line 117-139: “Blood typing was performed using the tube agglutination technique (TUBE) as previously described[23,35,36]. Briefly, to prepare a RBC suspension, 0.5 mL of EDTA-blood was centrifuged at 1,600 g for 10 minutes at room temperature. Plasma was collected into a separate tube and was stored at –20°C until tested for alloantibodies screening. The RBC pellet was washed by adding 2.5 mL of isotonic 0.9% NaCl solution. Following centrifugation at 1,000 g for 1 minute, the supernatant was removed, and the pellet was resuspended and then recentrifuged twice and finally reconstituted to a 5% RBC suspension. Polyclonal antibodies contained in type-B cat plasma (obtained from a type-B blood donor, collected with CPD anticoagulant at 1:7 ratio, stored at -20°C) were used as primary reagents for the detection of type A red cell antigens. Trititum vulgaris lectin (Sigma-Aldrich, 8 μg/mL) was used for the detection of type-B RBCs antigens as this lectin binds to the NeuAc terminal of the type-B ganglioside and therefore strongly agglutinates feline type-B RBCs, but does not agglutinate type-A RBCs [35]. A 0.9% NaCl solution (saline solution) was used as a negative control. In 3 glass tubes (PYREX® Tube Borosilicate glass 12x75mm, Coming Inc., NewYork USA), 50 μL of 5% RBC suspension was mixed with 100 μL of type-B plasma (anti-A reagent, tube A), 100 μL of Triticum vulgaris lectin solution (anti-B reagent, tube B), or 100 μL of saline (control reagent, tube C), respectively. These mixtures were incubated at room temperature for 15 minutes before centrifugation for 15 seconds at 1,000 g. Tubes were then gently agitated, and blood type was recorded for the tube where macroscopic agglutination was present (Figure 1).Type-B and AB samples were confirmed by back typing technique as previously described [12,37].
As samples collected in Sicily were tested after a minimum of 10 days from sampling, we preliminarily assessed the stability of samples by typing one sample of each blood type, stored at 4±2°C, using TUBE after 24, 48, 72, and 96 hours, and after 1, 2, 3, and 4 weeks of storage. Each sample stored at 4±2°C for up 1 month was correctly blood typed by TUBE.”
We provided a brief overview of the alloantibody screening as follow at lines 147-159: ”Alloantibody screening is a modified major crossmatch, using RBCs of a known blood group, thus screening for naturally alloantibodies against that group in the recipient. Alloantibody screening was performed as previously described on frozen plasma samples [37–39]. Briefly, in a glass tube25 μL of the type-A RBC suspension (5% in 0.9% NaCl) was added to 50 μL of type-B plasma. The same was done with type-B RBCs and type-A plasma and with type-A and type-B RBCs with type-AB plasma. The solution was gently mixed, incubated for 15 minutes at room temperature and then centrifuged for 15 seconds at 1,000 g. Agglutination was evaluated by gently agitating the tube to resuspend the non-agglutinating RBCs in the cell button. The presence of alloantibodies against RBCs of a different blood type was identified if macroscopic agglutination was present, as in the previous description of the tube technique for blood typing. In addition, where agglutination was not macroscopically visible, microscopic agglutination was evaluated at x40. Any RBC agglutination, macroscopic and/or microscopic, but not rouleaux presence alone, was considered a positive result for presence of alloantibodies in the tested plasma sample.”
Finally, as suggested purebred cats and cats with no reported breed have been removed and the revised manuscript now reports data from only non-pedigree DSH cats. In addition we checked the laboratory database again and found more data on breed for some samples with no breed reported before. Finally, we removed from the study the data from the 15 blood samples from Calabria region. Now you can find data from only Sicily and Lombardy regions
-- Section of results should be changed by removing the results from purebred cats since they are known to have different blood group frequencies and can alter results. In this study, the presence of a few purebred cats does not give us more information (indeed, probably removing them the statistical differences between northern and southern samples could increase). Moreover, they are almost all from northern Italy. In table 2 it is not specified how many blood groups were obtained from each breed.
The group without breed specification has been added in DSH group? Please clarify and think to exclude these data.
R: as suggested and previously stated we have deleted all the data relating to purebred cats, cats with no reported breed and cats from Calabria. The study now focuses only on non-pedigree cats from Sicily and Lombardy regions.
Specific Comments:
Summary
Line 24: change “useful” with “ usefulness”
R: done
Abstract:
Change the total number of samples and prevalences excluding purebred cats and cats from Calebria Region. Cats without breed specification? Should be excluded or separated.
R: as suggested and previously stated we have deleted all the data relating to purebred cats, cats with no reported breed and cats from Calabria. The study now focuses only on non-pedigree cats from Sicily and Lombardy region
Introduction
Line 44 The most clinically important
R: done
Lines 46-47 It is not clear, please change
R: changed as follows at line 48-49: ”Cats have antibodies against red blood cells (RBCs) of different blood types, called alloantibodies.”
Line 56 please change with type A and AB kittens and delate from “Kittens …..to …..RBCs”
R: done as suggested
Line 77 I suggest deleting Calebria Region, since the sample number from this region probably (you have not reported the number but t is possible to extrapolate it) is very low (15-16 samples)
R: we have followed the suggestion and removed from the study the data of the 15 blood samples from Calabria region. Now you can find data from only Sicily and Lombardy regions
Line 80 please, change with …from the Southern region (Sicily)….compared to cats from the northern region (Lumbardia region)
R: done and we tried to highlight this information as much as possible in the manuscript
Material and Methods
Line 87: 3.3 is probably a repeated error
R: yes it is an error and we have deleted it
Samples and population
Lines 91-92 : the inclusion of potentially sick cats could affect the results of the study. Results of blood typing can be affected by disease states of the animals. Particularly FeLV, anemia, and autoagglutination may contribute to the test inaccuracies as shown by Seth et al. 2011. Could you describe the diseases affecting those sick cats included in the study? Could you include the number of sick cats and the type of disease (especially if FeLV-anemia and autoagglutinations occurred) in the results section and Table1? Could you include in the discussion this possible limitation? Did you perform FeLV test in all cats?
R: unfortunately we have no information on the health status of the cats we included in this study. We used a blood typing method, the tube agglutination, in which results are not affected by autoagglutination, which was eliminated by washing RBCs,or anemia which was eliminated by the preparation of fixed-concentration RBC suspensions to remove the effect of Hct, which is particularly useful in anemic samples.
We know that rare blood types such as type-B or AB should be confirmed by use of laboratory techniques and alloantibody screening and in this study we used the alloantibody screening to confirm type-B and AB blood samples. Finally, please remember that FeLV is absent in the feline population in Sicily, therefore it is very unlikely that blood typing results were affected by this infectious disease.
Also, please, specify how long the samples have undergone to storage. Were the blood groups carried out immediately after collection with the tube test or stored? Since the analysis was conducted in northern Italy, the samples sent from southern Italy were refrigerated? Has it been used Frozen plasma?). The storage time and method used may have affected some results of blood typing due to stability related problems of samples during the time
R: the samples were stored at 4-6°C until analysis that was all performed at Milan University, in the Lombardy region, where the samples of northern cats were recruited. The samples sent from Southern Italy were refrigerated during shipment and until analysis, and alloantibody screening was done on frozen plasma samples at -20°C within 6 months of RBC separation.
You can find the information on how long the samples were stored as follows at lines 106-109 in the M&M section”All analyses were performed at the Veterinary Transfusion Research Laboratory (REVLab) of the University of Milan where the Northern Italy cats were sampled. Blood samples from cats in Southern Italy were stored at 4-6°C after collection and were shipped refrigerated to the REVLab where they were analyzed within 48 hours of arrival.”
And in the Results section at lines 207-208 as follow: “Alloantibodies screening were performed on frozen plasma samples within six months from RBCs separation.”
For the alloantibody screening and, in particular for alloantibody screening in type-A blood samples, the long storage time of frozen samples may have affected some results and we added this as a limitation in discussion section as follows at line 278-281:”In addition, it is possible that the delay in separating plasma from RBCs in some samples, and the long storage time before analysis, could have influenced these results, reducing, and thus limiting, recognition of the weak RBC agglutination reactions characteristic of type-A alloantibodies.”
Line 98: delete the phrase “This study…..only”.
R: done
TUBE test
Line 103-108: please rewrite and describe deeply the TUBE method used with reagents and all the methodological steps. Do you have any image of the results?
R: we added the description of the TUBE method and the image of the three different results.
You can find this new information in the M&M section as follows at lines 117-139: “Blood typing was performed using the tube agglutination technique (TUBE) as previously described [23,35,36]. Briefly, to prepare a RBC suspension, 0.5 mL of EDTA-blood was centrifuged at 1,600 g for 10 minutes at room temperature. Plasma was collected into a separate tube and was stored at –20°C until tested for alloantibodies screening. The RBC pellet was washed by adding 2.5 mL of isotonic 0.9% NaCl solution. Following centrifugation at 1,000 g for 1 minute, the supernatant was removed, and the pellet was resuspended and then recentrifuged twice and finally reconstituted to a 5% RBC suspension. Polyclonal antibodies contained in type-B cat plasma (obtained from a type-B blood donor, collected with CPD anticoagulant at 1:7 ratio, stored at -20°C) were used as primary reagents for the detection of type A red cell antigens. Trititum vulgaris lectin (Sigma-Aldrich, 8 μg/mL) was used for the detection of type-B RBCs antigens as this lectin binds to the NeuAc terminal of the type-B ganglioside and therefore strongly agglutinates feline type-B RBCs, but does not agglutinate type-A RBCs [35]. A 0.9% NaCl solution (saline solution) was used as a negative control. In 3 glass tubes (PYREX® Tube Borosilicate glass 12x75mm, Coming Inc., NewYork USA), 50 μL of 5% RBC suspension was mixed with 100 μL of type-B plasma (anti-A reagent, tube A), 100 μL of Triticum vulgaris lectin solution (anti-B reagent, tube B), or 100 μL of saline (control reagent, tube C), respectively. These mixtures were incubated at room temperature for 15 minutes before centrifugation for 15 seconds at 1,000 g. Tubes were then gently agitated, and blood type was recorded for the tube where macroscopic agglutination was present (Figure 1).Type-B and AB samples were confirmed by back typing technique as previously described [12,37].
As samples collected in Sicily were tested after a minimum of 10 days from sampling, we preliminarily assessed the stability of samples by typing one sample of each blood type, stored at 4±2°C, using TUBE after 24, 48, 72, and 96 hours, and after 1, 2, 3, and 4 weeks of storage. Each sample stored at 4±2°C for up 1 month was correctly blood typed by TUBE.”
Probably you decided to use the Tube agglutination test since it is considered the original gold standard method for blood typing and alloantibody screening based on 100% sensitivity and specificity for detection of the A antigen. Nevertheless, this method has been demonstrated to have an excellent concordance of typing results by the use of other more rapid and objective techniques using monoclonal antibodies. Did you use the rapid and accurate immunochromatographic test kits commercially available to compare and confirm blood groups? Tube test could induce errors and inaccuracies in case of autoagglutination, severe anemia, and or FeLV+ cats since the interpretation of results is subjective and sometimes the presence of alloantibodies could be strongly reduced in these sick cats. In some of these cases, the immunochromatographic test kits could be more objective to detect blood group, especially in the case of diseased cats.
R: as stated before we used back typing technique and alloantibody screening to confirm the rare B and AB blood type. With tube agglutination results are not affected by autoagglutination, which was eliminated by washing of RBCs,or anemia which was eliminated by the preparation of fixed-concentration RBC suspensions removing the effect of Hct, particulalyr useful in anemic samples. In addition, in each tested sample one column is dedicated for the autocontrol to check for possible problems such as persistent autoagglutination. All these steps and the presence of the control render this method very sensitive and specific even when blood typing autoagglutinated or anemic samples. Finally, please remember that FeLV infection is not present in cats from Sicily region. For these reasons we did not use any additional techniques to confirm our results, nor the accurate immunochromatographic test kits commercially available (with which we are very familiar because we have evaluated these kits in the past in the paper SPADA E et al Evaluation of an immunochromatographic test for feline AB system blood typing. Journal of Veterinary Emergency and Critical Care. 2016;26(1):137-141. DOI:10.1111/vec.12360). We usually use this test to type cats in emergency situations.
What do you mean for “we evaluated some data….literature”? Please rewrite.
R: as suggested by another reviewer we deleted this sentence. However we performed this evaluation becausewhen we decided to start this study we needed to know how long we could store the samples before blood typing. This is because mail between Sicily island and the rest of Italy can take several days and sometimes long delays can occur due to the bad sea conditions. However, we didn’t found any literature/published data about length of time of sample storage at 4-6°C before typing with TUBE.
We rewrote the sentence as follow at lines 106-109 in the M&M section: ”All analyses were performed at the Veterinary Transfusion Research Laboratory (REVLab) of the University of Milan where the Northern Italy cats were sampled. Blood samples from cats in Southern Italy were stored at 4-6°C after collection and were shipped refrigerated to the REVLab where they were analyzed within 48 hours of arrival.”
Line 104: please, write “and” a microscopic evaluation is necessary
R: done
LINE 131 “as per” is an error? Please, add citations
R: as suggested by another reviewer, we have deleted all the calculations about major and minor transfusion reactions and neonatal isoerythrolysis as these calculations were very theoretical - assuming that that recipient cats come from the same population as donors and also that matings are random from these populations which is very unlikely.
Line 153 how many samples were analyzed from Sicily and how many from Calebria? Unclear data. It is possible to extrapolate a very low number of samples from Calabria (15-16) and I suggest to remove them.
R: we have followed the suggestion and removed from the study the data from the 15 blood samples from Calabria region. Now you can find data from only Sicily and Lombardy regions
LINES 157-161: purebred cats (56 and the majority from Northern Italy) and cats with no breed reported (41) should be excluded since each breed tends to have a specific prevalence of blood groups and this could distort the epidemiological data. Moreover, each breed has a low number of samples and the Ragdoll cats (the pure breed mostly represented, only from Northern Italy), are potentially AB and B types.
If you want to include these cats, please add in table 2 the results of blood typing dividing into 3 groups: each breed, cats with no breed data, and DSH. It allows to better evaluate the distribution of blood groups
R: as suggested purebred cats and cats with no breed reported have been removed and the revised manuscript reports data from only non-pedigree DSH cats from Sicily and Lombardy regions. In addition we checked the laboratory database again and found more data on breed for some samples with no breed reported before.
In the text (153-166) and in table 1 is not clearly reported the number of DSH cats and purebred cats
R: purebred cats have been removed and the revised manuscript reports data from only non-pedigree DSH cats from Sicily and Lombardy regions
In table 1 cats with no breed, data are included in DSH?
R: based on another reviewer’s suggestion Table 1 has been deleted and all information has been moved to the results section. Cats with no breed data have been excluded form the study. Now you can find only data from non-pedigree DSH cats from Sicily and Lombardy regions.
LINE 176-185 : I suggest to recalculate the percentage excluding purebred cats, cats from Calebria and cats with no breed data
R: we have done as suggested and we have excluded purebred cats, cats from Calabria and cats with no breed data.
LINE 194-196: delete. Obviously data are not valid due to the low number of samples from purebred cats, (the majority from Northern Italy).
R: deleted as suggested
LINES 197-200 and table 3: please, delete data from Calebria region since the number of samples are too low
R: we have deleted samples from Calabria as suggested. Now only samples of cats from Sicily and Lombardy are included in the study
LINE 214: please, specify how many type A, type B, and type AB cats have been tested for alloantibodies.
R: we added this information as follows at lines 208-211: “Due to insufficient residual volumes of plasma samples, alloantibodies were tested in only 327/448 (72.9%) samples: 196/233 (84.1%) from Northern Italy and 131/215 (60.9%) from Southern Italy (Table 3). In particular alloantibodies were tested in 268/378 (70.9%) type-A samples, 34/38 (89.5%) type-B samples and 25/32 (78.1%) type-AB samples.”
DISCUSSION
Please, reshape the discussion by removing at least the data from purebred cats and those from Calebria. These groups are too small and do not give useful valuable informations for discussion.
R: we have followed the suggestion and used only data from non-pedigree DSH cats and cats from Lombardy and Sicily regions for this study. The discussion was reshaped accordingly.
When you write “Southern and Northern Italy”, I suggest specifying that they are regions.
R: we added this specification multiple times in the manuscript
Line 263: this sentence is questionable. The number of individual groups (DSH, purebred cats) from the individual regions is too low and the sample is not representative of the entire population of cats in the considered areas. Furthermore, it is possible that the use of different blood typing kits or tests applied in the previous studies could induce differences in the interpretation and comparison of results.
R: we have deleted this sentence
Lines 264-265: Are the percentage you described obtained excluding purebred cats? Please, clarify.
R: as previously stated, now you can find only percentages relating to non-pedigree DSH cats from Lombardy and Sicily
Lines 276-279: Could you better explain this concept? Please, add citations and clarify
R: based also on the other reviewers’ suggestions we have modified this concept as follows at lines 253-258: “The high prevalence of type-AB cats in the Southern Italy cat population may reflect the relatively small gene pool from which this population has developed. Sicily is separated from the mainland by a marine natural barrier that prevents free movement of terrestrial fauna and the non-pedigree cat population is isolated from neighboring areas. These circumstances may have enhanced inbreeding in Sicilian domestic cats as is recognized in wild cats [50].”
Lines 286-287: from “where….. to…. 9.7%” please, delate.
R: deleted
Lines 287-290: this statement can’t be demonstrated. Cats with AB blood group are DHS cats? Purebreed cats?
R: we have deleted this sentence as the statement seems highly speculative
Line 300: What do you mean with the sentence “the first time stability of samples”?
R: when we started this study we needed to know the maximum length of storage of samples before blood typing. This is because mail between Sicily and the rest of Italy can take several days and sometimes delays can occur due to the bad sea conditions. However, we found no literature on practical storage times before typing with Tube.
We changed the sentence as follow in the M&M section at lines 136-139: “As samples collected in Sicily were shipped refrigerated to northern laboratory to be tested, we preliminarily assessed the stability of samples by typing one sample of each blood type, stored at 4±2°C, using TUBE after 24, 48, 72, and 96 hours, and after 1, 2, 3, and 4 weeks of storage. Each sample stored at 4±2°C for up 1 month was correctly blood typed by TUBE.”
LINES 300-304: Did you stored blood EDTA samples for 1 month before doing the analysis? It is not explained in the material and methods sections. Is the stability of samples maintained? Could you add some bibliography concerning this method and the stability of samples (RBCs antigens and alloantibodies)
R: to the best of our knowledge, no data are available on stability of samples to be typed with Tube, for this reason we performed the study on sample stability in our study. We added some of these information in the M&M section at lines 136-139 as follow. “As samples collected in Sicily were shipped refrigerated to northern laboratory to be tested, we preliminarily assessed the stability of samples by typing one sample of each blood type, stored at 4±2°C, using TUBE after 24, 48, 72, and 96 hours, and after 1, 2, 3, and 4 weeks of storage. Each sample stored at 4±2°C for up 1 month was correctly blood typed by TUBE.
In the Results section at lines 175-176: “Samples were blood typed after a mean time of 12 days (±SD 6 days, range 2-28 days).”
In Results section at lines 207-208: “Alloantibodies screening were performed on frozen plasma samples within six months from RBCs separation.”
Line 320: 12 weeks or 3 months of age?
R: we changed 12 weeks to 3 months of age
Lines 347-349: it is not specified the number of A, B, and AB cats in the purebred cats.
R: we deleted all the data relating to purebred cats. Now you can find only percentage relating to non-pedigree DSH cats from Lombardy and Sicily
Lines 348-349: “Due to the limited….determined” What samples do you mean?
R: we referred to residual volume after separation of ETDA blood, the plasma samples we used for alloantibodies screening. We have rewritten the sentence as follows at lines 295-296: “Due to the limited quantity of plasma samples, alloantibody screening was not always possible.”
Conclusions
Line 360: please, change with “a Region of Southern Italy….had a higher……and a lower prevalence…..compared to a Region of Northern Italy”
R: Done
Round 2
Reviewer 1 Report
Thank you for your responses.
Author Response
We thank the reviewer again for improving our manuscript
Reviewer 2 Report
The authors did a nice job responding to this reviewer's concerns, especially in removing pedigreed cats from the study.
Only a few comments are provided.
- Title. Move (Sicily) before Italy.
- Line 15. Add “the” after the word “against.”
- Line 56. Remove word “acute.” Although the antigen-antibody reaction is rapid, the removal of cells takes some time. Consequently, I don’t think of these as acute reactions.
- Line 57. Make two sentences because in one case the alloantibody is in the recipient’s plasma and in the second case it is in the donor’s plasma. Place a period after …. AB blood. Second sentence “Type-AB cats may have transfusion reactions when receiving type-B or type-A blood, due to anti-A or anti-B alloantibodies in the donor plasma.”
- Line 223. Move (Sicily) before Italy.
- Line 226 remove the word “the”
Author Response
Thank you again for your critical review and for the additional few minor comments to our manuscript. We have done all the changes (in red in the manuscript) as suggested in the 6 points below.
Reviewer 3 Report
Dear Authors,
thank you for your hard work. All revisions have been well organized and speculated. It seems a good job. I added some minor revisions:
"In addition we checked the laboratory database again and found more data on breed for some samples withIn addition we checked the laboratory database again and found more data on breed for some samples with no breed reported before no breed reported before"
please remove these data, the original data are enough
All these steps and the presence of the control render this method very sensitive and specific even when blood typing autoagglutinated or anemic samples. Finally, please remember that FeLV infection is not present in cats from Sicily region.
That is quite a harsh statement, is there a program of eradication or a screening program for cats in this Region? Do no cats come in this region from other Italian regions or countries? I suggest avoiding to write this sentence in the text. The use of biomolecular methods should be recommended to identify proviral FeLV DNA, considering that rapid tests are not definitively discriminatory and Cats could be positive for FeLV provirus only when using the most sensitive PCR protocol (Marenzoni ML et al. Comparison of three blood transfusion guidelines applied to 31 feline donors to minimise the risk of transfusion-transmissible infections. J Feline Med Surg. 2018;20(8):663-673. doi:10.1177/1098612X17727233).
Lines 71 and 227: please add this citation "Spada E, Antognoni MT, Proverbio D, Ferro E, Mangili V, Miglio A. Haematological and biochemical reference intervals in adult Maine Coon cat blood donors. J Feline Med Surg. 2015;17(12):1020-1027. doi:10.1177/1098612X14567549"
Line 224: please add this citation "Marenzoni ML et al. Comparison of three blood transfusion guidelines applied to 31 feline donors to minimise the risk of transfusion-transmissible infections. J Feline Med Surg. 2018;20(8):663-673. doi:10.1177/1098612X17727233"
From line 255 to line 263 Please add the immunochromatographic method and its sensitivity and specificity
Author Response
Dear Authors,
thank you for your hard work. All revisions have been well organized and speculated. It seems a good job. I added some minor revisions:
"In addition we checked the laboratory database again and found more data on breed for some samples withIn addition we checked the laboratory database again and found more data on breed for some samples with no breed reported before no breed reported before"
please remove these data, the original data are enough.
R: The new statistical analysis in the revised manuscript is based solely on the data from DSH cats. We would appreciate it if you could accept all the data reported in the revised version of the manuscript as this is a complete population that reaches the number of cases required for the statistical significance of this epidemiological study
All these steps and the presence of the control render this method very sensitive and specific even when blood typing autoagglutinated or anemic samples. Finally, please remember that FeLV infection is not present in cats from Sicily region.
That is quite a harsh statement, is there a program of eradication or a screening program for cats in this Region? Do no cats come in this region from other Italian regions or countries? I suggest avoiding to write this sentence in the text. The use of biomolecular methods should be recommended to identify proviral FeLV DNA, considering that rapid tests are not definitively discriminatory and Cats could be positive for FeLV provirus only when using the most sensitive PCR protocol (Marenzoni ML et al. Comparison of three blood transfusion guidelines applied to 31 feline donors to minimise the risk of transfusion-transmissible infections. J Feline Med Surg. 2018;20(8):663-673. doi:10.1177/1098612X17727233).
R: you are right, there isn’t a program of eradication or a screening program for cats in Sicily. This statement was based on epidemiological studies and it is possible that cats move among regions. Therefore, as suggested, we avoided commenting on FeLV infection in Sicily in the manuscript
Lines 71 and 227: please add this citation "Spada E, Antognoni MT, Proverbio D, Ferro E, Mangili V, Miglio A. Haematological and biochemical reference intervals in adult Maine Coon cat blood donors. J Feline Med Surg. 2015;17(12):1020-1027. doi:10.1177/1098612X14567549"
R: as suggested we added this reference at line 71 and 227
Line 224: please add this citation "Marenzoni ML et al. Comparison of three blood transfusion guidelines applied to 31 feline donors to minimise the risk of transfusion-transmissible infections. J Feline Med Surg. 2018;20(8):663-673. doi:10.1177/1098612X17727233"
R: as suggested we added this reference at line 224
From line 255 to line 263 Please add the immunochromatographic method and its sensitivity and specificity
R: we have added this information as follows at lines 268-270: ”There is less misclassification in more recent studies using the immunochromatographic method which has a higher sensitivity and specificity than the card test for typing both type-B and AB samples [53]
The new reference is:
- Spada, E.; Proverbio, D.; Baggiani, L.; Bagnagatti De Giorgi, G.; Perego, R.; Ferro, E. Evaluation of an immunochromatographic test for feline AB system blood typing. J Vet Emerg Crit Care 2016, 26, 137–141.